# Collision Cross Section Prediction with Molecular Fingerprint Using Machine Learning

**DOI:** 10.3390/molecules27196424

**Published:** 2022-09-29

**Authors:** Fan Yang, Denice van Herwerden, Hugues Preud’homme, Saer Samanipour

**Affiliations:** 1Institut des Sciences Analytiques et de Physico-Chimie Pour l’Environnement et les Materiaux (IPREM-UMR5254), E2S UPPA, CNRS, 64000 Pau, France; 2Van ’t Hoff Institute for Molecular Sciences (HIMS), University of Amsterdam, Science Park 904, 1098 XH Amsterdam, The Netherlands; 3UvA Data Science Center, University of Amsterdam, 1098 XH Amsterdam, The Netherlands

**Keywords:** collision cross section, ion mobility spectrometry, non-target screening, machine learning

## Abstract

High-resolution mass spectrometry is a promising technique in non-target screening (NTS) to monitor contaminants of emerging concern in complex samples. Current chemical identification strategies in NTS experiments typically depend on spectral libraries, chemical databases, and in silico fragmentation tools. However, small molecule identification remains challenging due to the lack of orthogonal sources of information (e.g., unique fragments). Collision cross section (CCS) values measured by ion mobility spectrometry (IMS) offer an additional identification dimension to increase the confidence level. Thanks to the advances in analytical instrumentation, an increasing application of IMS hybrid with high-resolution mass spectrometry (HRMS) in NTS has been reported in the recent decades. Several CCS prediction tools have been developed. However, limited CCS prediction methods were based on a large scale of chemical classes and cross-platform CCS measurements. We successfully developed two prediction models using a random forest machine learning algorithm. One of the approaches was based on chemicals’ super classes; the other model was direct CCS prediction using molecular fingerprint. Over 13,324 CCS values from six different laboratories and PubChem using a variety of ion-mobility separation techniques were used for training and testing the models. The test accuracy for all the prediction models was over 0.85, and the median of relative residual was around 2.2%. The models can be applied to different IMS platforms to eliminate false positives in small molecule identification.

## 1. Introduction

A large number of chemicals have been released into the environment by human activities, such as agriculture, industrial productions, and their relative byproducts. Once these chemicals enter the environment, transformation products (TPs) can be produced through hydrolysis, photosynthesis, and biological metabolism [1,2,3,4,5,6]. Most of these chemicals and their TPs are missing molecular and/or structure information. Thus, these chemicals’ human and environmental risk assessments remain an open question [6,7,8,9,10,11,12]. Although most legacy pollutants have been banned for decades in many countries, they can still be detected at trace-level in the environment [2,13,14,15]. The known pollution is only the tip of the iceberg compared to the number of environmental hazards [1,13,14].

Non-target screening/analysis (NTS) is considered as an appropriate methodology to identify a variety of chemicals, especially for the unknown unknowns, such as contaminants of emerging concern (CECs) [16,17,18]. High-resolution mass spectrometry (HRMS) coupled with gas or liquid chromatography (GC or LC) is the most commonly used analytical technique in human health and environmental assessments. Thanks to the advance of HRMS, it has been increasingly applied in NTS studies in the last decades [17,19,20,21]. HRMS (i.e., Time-of-flight (TOF) and Orbitrap) maintains a high mass accuracy within ±5 mDa *m*/*z* error, and it can be acquired in full scan MS data or plus MS/MS data [10,21,22,23,24]. The accurate mass of the parent ion and the fragments are used to identify unknowns [17,19,21]. The isotopic pattern is one of the additional criteria which can help determine the presence of hetero-elements in non-target analysis [25]. However, mass spectral information is not enough for highly confident structural elucidation [22,25,26]. Therefore, inclusion of orthogonal sources of information such as measured or predicted retention time and/or retention time indices is necessary [21,27,28]. Such measurements are complex to perform and require particular experimental conditions [29,30,31].

Collision cross section (CCS) is a platform-independent measure of chemical structure in the gas phase and the three-dimensional space [32,33,34]. Studies have demonstrated that the inter-laboratory CCS biases are within 2% for the same IMS technique [35,36]. Moreover, cross-platform biases are below 3% for over 98% of the chemicals included in their studies [37,38]. Drift tube ion mobility (DTIM) and traveling wave ion mobility (TWIM) are two of the most used IMS techniques to measure the CCS value or drift time [37,39]. CCS value and drift time have been employed in NTS as an additional source of information to increase confidence level in structural elucidation [40,41,42]. In addition to experimentally defined CCS values, CCS values can be estimated/predicted via theoretical calculations or Machine Learning (ML) [43,44]. ML CCS predictions take advantage of large datasets of the experimentally defined CCS values to train, validate, and test the regression models [44]. Zhou et al. [45] reported the first CCS prediction tool using the support vector regression (SVR) ML algorithm for metabolites. Plante et al. [46] published a deep neural networks CCS prediction strategy for cross-platform CCS measurement. The currently available CCS prediction tools rely on molecular descriptors or the combination of the chemical class and the *m*/*z* value of the parent compound [44,45,46,47,48,49,50,51,52]. Molecular fingerprints, which are more accurate and representative of the structure of a molecule [53], have not been used for the prediction of CCS values due to the difficulties associated with variable selection.

This study proposes a novel approach for CCS prediction using molecular topology fingerprints instead of molecular descriptors. First, we built a classification model to predict the chemical super classes based on their fingerprints. This model was used to classify chemical super classes. Then, CCS prediction models were developed for each super class. Additionally, all 13,324 chemicals were combined and to build a direct CCS prediction model. We also evaluated the impact of the chemical classes on the model accuracy.

## 2. Materials and Methods

### 2.1. Datasets

Experimental CCS databases and chemical information were collected from Zenodo, PubChem, and published articles as referenced in Table 1. Firstly, we retrieved all the missing SMILES notations from PubChem by PubChem CID using the Python PubChemPy library [54]. All the datasets were concatenated, and molecular fingerprints were generated by RDKit [55] (Open-source cheminformatics https://www.rdkit.org) (accessed on 10 April 2022) modules in Python. Hence, a dataset containing PubChem CID, SMILES [56,57], and empirical CCS value was saved as a csv file ready for model development and validation. The datasets and the source codes are available at https://github.com/fyang22/CCS-Prediction-Publish (accessed on 10 April 2022). Additional details about model optimization and construction are available in the Appendix A.

The merged dataset included 13,324 unique empirical CCS values from 108.4 to 450.6 Å${}^{2}$, measured by TWIM and DTIM. The merged dataset of 3313 chemicals was categorized into 43 super classes, including POPs, lipids, sugars, metabolites, hormones, drugs, etc. This dataset was then used for a classification model training and testing. Topological torsion (TT) fingerprints were chosen as features to encode chemical structure. TT fingerprints were first introduced by Nilakantan et al. [58], which describe the atom type, the topological distance between two atoms within four bonds, and torsion angles [59]. Four examples of molecular substructures are shown in Figure 1. The SMILES were converted to 1024 bit-strings fingerprints (FPs) by the implemented module in RDKit. The FPs were used to calculate molecular similarity, then visualized by principal component analysis (PCA) and fit machine learning models.

### 2.2. Overall Workflow

This study consists of two major parts and three models, and the workflow is summarized in Figure 2. Firstly, we developed a classification model to categorize chemicals into five groups, so-called “super class”, based on their FPs similarity. The number of the “super class” was selected to create a balanced distribution of chemicals in each class. Five class-based CCS prediction models were developed using the optimized predicted category. Meanwhile, a direct CCS prediction model was built with the complete dataset without considering chemical categories. We also compared the two strategies to assess the prediction accuracy of these two modeling approaches. Finally, we applied the models to NORMAN SusDat (i.e., 101,684 chemicals) and carried out the direct and class-based prediction of the CCS values for SusDat.

#### 2.2.1. Dataset for Classification Model

The dataset consisted of the identified chemical super classes which were merged from three CCS libraries [60,61,62]. This split dataset was used for chemical classification model training, validation, and testing. Initially, 43 super classes were defined, where most super classes contained less than 20 chemicals. To avoid overfitting of the classification model, we merged different super classes based on the calculated similarity scores of the chemicals. This enabled a more balanced distribution of chemicals in each super class. First, we calculated pair-wised fingerprint similarity by the Tanimoto similarity using RDKit. Tanimoto coefficient is a way to calculate the distance metric using molecular fingerprints [53,63]. Based on the distribution of the chemicals, super classes, and the similarity scores (plotted in Figure 3a), we kept the 5 super classes with the highest population of chemicals (listed in Figure 3b) and used them as ground truth. Chemicals in other super classes were assigned to one of the referred classes based on their similarity with a minimum similarity threshold of 0.6 since around 97% of pair-wise similarities were under 0.6 (shown in Figure 3a). Chemicals (n = 118) not meeting the similarity score criteria were manually assigned to a new super class (5 super classes) based on their characterized functional groups. Meanwhile, we kept the chemicals from the same given class (43 super classes from the raw dataset) in the same new super class. The final dataset consisted of 5 super classes having around 1000 unique chemicals in each class (in Appendix A), the classification of chemicals is visualized in Figure 3b. This dataset was used for random forest classification modeling. The final dataset for classification included fingerprints with 1024 bit-strings and the assigned super classes. Our super-class reassignment strategy effectively differentiated chemical classes from each other. For example, Organic acid and derivatives (in blue) and Benzenoid (in green) are two separate clusters in the middle left and in the bottom left.

#### 2.2.2. Dataset for Regression

For CCS regression modeling, we only considered protonated ions (8620 chemicals of [M + H]^+^), deprotonated ions (4589 chemicals of [M − H]^−^) and radical ions (115 chemicals of [M]^.^). Then, all the replications were removed by the SMILES, adduct ion and CCS values. Meanwhile, we calculated the standard deviation of CCS values for the same chemicals (same SMILES and adduct ion). In the training and test datasets, 642 chemicals have replications with different measured CCS values. The median of relative standard deviation (RSD) was about 1.4% (shown in Appendix A) for both positive and negative ionization mode, and studies from multiple laboratories, which are consistent with the results reported by Hinnenkamp et al. [37] and Feuerstein et al. [38] Aspartame resulted in RSD of 12.5%, Picache et al. [60] recorded a CCS value of 127.4 Å${}^{2}$ for Aspartame [M + H]^+^, which is 40 Å${}^{2}$ lower than the one measured in other references. Different Aspartame CCS values are also recorded in https://pubchem.ncbi.nlm.nih.gov/compound/134601#section=Collision-Cross-Section (accessed on 1 June 2022). Hence, this dataset, collected from different laboratories and measured by different IM-MS platforms, was appropriate for CCS prediction. The entire dataset contained 13,324 unique empirical CCS values ranging from 108.4 to 450.6 Å${}^{2}$, covering metabolites, drugs, lipids, etc., and it is available in Appendix A.

**Table 1 molecules-27-06424-t001:** Summary of the dataset used in CCS prediction model optimization.

Reference	Number of Chemicals	Instrument *
Picache et al. [60]	1195	Agilent 6560 IM-QTOF MS
Hines et al. [64]	1304	Waters Synapt G2-Si HDMS
Celma et al. [40]	631	Waters VION IMS-QTOF MS
Zheng et al. [61,62]	891	Agilent 6560 IM-QTOF MS
Belova et al. [65]	145	Agilent 6560 IM-QTOF MS
Bijlsma et al. [51]	193	Waters VION IMS-QTOF MS
PubChem [66]	8965	

* Agilent: Drift tube ion mobility (DTIM), Waters: Traveling wave ion mobility (TWIM).

### 2.3. Modeling

In this study, we optimized three models: (1) Class prediction, (2) Class-based CCS regression model, and (3) a direct CCS regression model. A super class prediction model was first optimized using random forest classification. This model was used to assign the super class (i.e., five classes) of the whole dataset. Then, a regression model was built for each super class to predict the CCS values based on the FPs. Finally, we developed a model using only molecular FPs for CCS prediction. We compared the pros and cons of two CCS prediction approaches. All the modelings were performed using a 5-fold cross-validation by GridSearchCV build-in functionality in Scikit-learn. The details of each modeling strategy are provided below.

#### 2.3.1. Class Prediction

The Class prediction model was first optimized using the random forest classification algorithm. The dataset was split into a training set (80%, n = 836) and a test set (20%, n = 210) with even distribution by super classes. In the random forest classifier, different hyper-parameters impact the model accuracy differently [67]. In this study, we focused on the number of trees in the random forest (n_estimators) and the minimum number of samples required at each leaf node (min_samples_leaf). These two parameters appeared to have the highest impact on the balance between the model robustness and accuracy. We generated a grid with 25 candidates for the number of trees ranging from 100 to 200 and 2 to 15 for minimum sample leaf. For each model, we performed 5 folds of cross-validation to assess the model accuracy. The model with the highest cross-validation accuracy was chosen as the optimized classification model, and the GridSearchCV scores are plotted in Appendix A. The accuracy and F1 scores of each class are listed in Table 2.

#### 2.3.2. Class-Based CCS Regression

For class-based regression modeling, we applied the optimized classification model (mentioned above) to the entire dataset, and the results are shown in Appendix A. We independently performed the CCS prediction modeling for 5 data splits based on this classification, using the random forest regression algorithm. A total of 80% of the datasets were trained and tested by the rest. Similarly, we generated a grid with 50 candidates and the number of tree fits of 100 to 500. To avoid overfitting, the minimum sample leaf was set from 5 to 20. For each model and each class, 5 folds of cross-validation were evaluated to assess the model accuracy (Appendix A).

#### 2.3.3. Direct CCS Regression

For comparison, we developed and tested a direct CCS prediction model for the entire dataset (13,324 compounds). A total of 80% of the data was used to train the model, and 20% of the data to test with 5-fold cross-validation (Appendix A). Similarly to the class-based CCS prediction model, n_estimators, and min_samples_leaf were optimized. The hyper-parameter optimization followed the same steps as class-based modeling (mentioned above). The model details and accuracy are listed in Table 3.

## 3. Results

### 3.1. Random Forest Classifier and Regression Prediction Model

Random forest is a suitable supervised machine learning algorithm for categorical and nonlinear data. We used a random forest classifier model to divide chemicals into 5 super classes by their molecular fingerprints. Then, we developed two CCS prediction strategies using molecular fingerprints. One is based on molecular super classes and molecular fingerprints, and another is a direct prediction by molecular fingerprints. As a CCS value is related to the chemical structure, we described each chemical structure by 1024 bit-strings molecular fingerprints, which were used as the prediction features. Each bit represents a substructure of a chemical, and some refer to a characteristic chemical substructure. These bits build up sets of nodes and leaves, then a decision tree.

A collection of decision trees results in a random forest model (decision trees files are available in Appendix A). In order to obtain a generalized CCS prediction model, we merged 7 CCS libraries containing 13,324 unique CCS values (108.4 to 450.6 Å${}^{2}$) measured by TWIM and DTIM platforms from multiple laboratories. Additionally, using a merged dataset for modeling allowed us to understand the variation of CCS measurement.

### 3.2. Evaluation of Classification Model

We obtained a classification model to separate 5 super classes with a global test accuracy (R${}^{2}$) ≥ 0.871. In the classification model, it is crucial to have sufficient examples and similar training weights for each class. For example, if the dataset is randomly split to 80% of the training set that contains 50 organoheterocyclic compounds but over 200 chemicals of other classes, it would lead to insufficient training for organoheterocylic compounds and an overfitting problem, which can impact the overall performance of the classifier prediction. As shown in Table 2, the training and test sets were evenly distributed by super classes before modeling. The F1 score was over 0.9 for two classes and over 0.82 for the other three, indicating that the training data were balanced between classes. To further evaluate the classification model, we also generated a confusion matrix (Figure 4).

Our model correctly predicted the super class of around 87% of the chemicals while around 8% of Organic acids and derivatives were classified as Organoheterocyclic compounds or Organic oxygen compounds. We noticed that errors frequently occurred in carboxylic acid compounds with phosphate esters or peptides. We randomly selected 3 incorrectly classified chemicals in each class. For instance, sulfadimethoxine (Figure 5a) was defined as Benzenoids due to an aniline. Nevertheless, it also contains pyrimidine, which was predicted as an Organoheterocyclic compound. Similarly, 3-Methyloxindole (Figure 5b) is an oxinole derivative consisting of a benzene ring and a heterocyclic with nitrogen. It was assigned to Organoheterocyclic (indole) in the collected dataset but went to Benzenoids compounds by prediction. We further investigated these incorrect classifications by examining the feature importance, shown in Appendix A. Figure 1 shows a possible substructure of the most relevant bit-strings. For example, bit 792 (Figure 1b) would define whether a compound is classified as a Benzenoid or Organoheterocyclic compound. On the other hand, the bit-string 842 (Figure 1c) was used to decide whether a chemical should go to Organic oxygen compounds. None of the bit-strings displayed significant importance from others, indicating that the “incorrect” classification mainly has to do with which functional groups were given the higher priority when the original training set was being compiled.

### 3.3. Evaluation of Regression Models

In class-based modeling, the prediction R${}^{2}$ was from 0.860 to 0.933, and the median relative error (MRE) of prediction was from 1.89% to 2.33% (Table 3). Direct CCS prediction, on the other hand, reached an R${}^{2}$ of 0.95 and MRE of 2.2%, showing a good performance. Although we dropped replicated chemicals having the same CCS values before generating the modeling, considering that this dataset was merged by inter-laboratory studies, some chemicals might have been seen during training. Thus it can affect the prediction accuracy. Chemicals with less measurement deviation will increase the accuracy. On the contrary, those who have a significant deviation will bias prediction performance. We confirmed that for the direct prediction model, only 2% of the chemicals were common over 2665 test samples. The dataset was split by category in the class-based prediction, and the replications percentage was varied by chemical class. About 10% chemicals in the test set of Organic oxygen compounds were used in training before prediction, and less than 5% for other classes. Furthermore, except for a few outliers, the deviation of replications was under 6%. Therefore, we considered that the impact of replicated chemicals was negligible.

Additionally, we compared the performance of class-based models. Organic oxygen compound model obtained the lowest accuracy due to the lack of training data. Moreover, in its test split, the relative error ≥10% only occurred to macromolecules (e.g., maltodecaose ($C_{60}H_{102}O_{51}$)), contributing 15% to the test split, which resulted in poor prediction accuracy. Since we could not remeasure outliers’ CCS values, we hypothesize that the error is associated with the compact and complex chemical structure. For instance, IMS measures the rotational-average surface of the maltodecaose ion. While a 1024 bit fingerprint is not enough to represent its complex chemical structure, resulting in a relative prediction error of 41.9% (true CCS at 390.3 while predicted 226.6 Å${}^{2}$). Another possible reason can be the training weight. The dataset size of Organic oxygen compounds were almost 5 times less than Lipids and lipid-like molecules dataset, and glucose was the minority in the Organic oxygen compounds dataset. The model cannot properly generate the chemical rarely present during training. Therefore, higher accuracy was reached by Lipids and lipid-like molecules model and the direct prediction model. Outliers of other models were further investigated (shown in Appendix A), and Figure 6 and Figure 7a compare the predicted results of class-based models and direct prediction model. Four error cases have occurred to macromolecules (e.g., Diphenyl phosphate ($C_{39}H_{34}O_{8}P_{2}$)), which can be explained by the same hypothesis as maltodecaose (mentioned above). Metronidazole ($C_{6}H_{9}N_{3}O_{3}$) has 6 empirical CCS values measured with Waters TWIM, 5 were between 124 to 133 Å${}^{2}$, while 200 Å${}^{2}$ was measured by Picache et al. [60], leading a −61 Å${}^{2}$ residual error (predicted CCS = 139.3 Å${}^{2}$). L-tenuazonic acid ($C_{10}H_{15}NO_{3}$) was predicted to have a twice higher CCS than the measured one by the class-based model (35% higher by the direct prediction model). It might result from an inappropriate prediction by certain important features. Predicted CCS of vinyl acetate ($C_{4}H_{6}O_{2}$) was 127.4 Å${}^{2}$ through the class-based model, and 147.9 Å${}^{2}$ by direct prediction, while the empirical one was 227.2 Å${}^{2}$. We hypothesize that vinyl acetate might be polymerized leading to higher measured CCS values. Benefiting from datasets from multiple sources, class-based and direct prediction models can verify experimental CCS and evaluate the inter-laboratory and inter-platform deviation.

Figure 6 compares empirical and predicted CCS values generated by different models. We noticed that the direct prediction model was less biased by chemical class and structure, small and/or macro molecules, leading to higher prediction accuracy than the class-based prediction results. Although class-based models generated lower MREs (Table 3), a higher residual error was obtained in vinyl acetate and macro molecules resulting in lower R${}^{2}$. As we can see in Figure 7c, over 25% of the test dataset obtained relative residual lower than 1%, and class-based models gained slightly higher, at 26.6%. All prediction models were further evaluated by feature importance (shown in Appendix A). In both prediction approaches, the most relevant features divided chemicals into relative low CCS and high CCS. In other words, the decision tree was made of different CCS ranges based on certain substructures. For example, the most relevant feature in Organic acids and derivatives CCS prediction model was bit 588 (Figure 1d). If a chemical has its represented substructure, this chemical will be considered as CCS >150 Å${}^{2}$, which might yield the prediction error for l-tenuazonic acid. Overall, the direct CCS model generated the best prediction performance, and a more extensive dataset ensured a more robust model.

MetCCS was a support vector regression (SVR) based on a prediction method only for metabolites. It achieved an excellent R${}^{2}$>0.96 with the intra-laboratory and inter-laboratory measurements, relative residual was within 5% [45]. Bijlsma et al. [51] developed an artificial neural network (ANN)-based CCS prediction tool and [52] published an multivariate adaptive regression splines (MARS) CCS prediction model. Both were trained by TWIM data, and the relative error was within 6% for 95% of the chemicals. Belova et al. [68] compared experimental DTIM measured CCS values to predicted CCS values by the ANN-based and MARS-based predictors. A total of 95% of the protonated and deprotonated ions observed the relative error under 6.7%. However, only 56 compounds with 108 DTIM measured CCS values were compared in their study. We obtained comparable results by direct and class-based models, 87% of predicted results obtained the relative error within 7% (Figure 7c). DeepCCS is a more generalized CCS prediction model generated by SMILES with the deep neural network. R${}^{2}$ was greater than 0.97, and MRE was below 2.6% [46]. However, only 1637 datasets were initially used to train the model, and the prediction power might be declined by chemical class [49,50]. We achieved a comparable accuracy for a wider scope of chemicals by direct prediction model (R${}^{2}$ over 0.95, MRE within 2.2%). AllCCS and CCSbase generated better accuracy, with R${}^{2}$ over 0.98 and MRE below 2%, since both tools used a larger and more diverse training set than DeepCCS and MetCCS. More structural-related features were emphasized in their studies. Considering our models, we reached comparable MREs with other tools and over 90% of the chemicals predicted within 8% relative residual. The results are satisfied with the CCS measurement bias via different IMS instrumentation and techniques [37].

### 3.4. Application on SusDat

NORMAN SusDat database contains over 111,000 environmentally relevant chemicals, with SMILES, accurate mass, and physiochemical properties [69]. We applied direct CCS prediction and class-based CCS prediction to the SusDat database, which contains chemicals that have never been seen during training and test, such as antibiotics and transformation products. A total of 96 % of the chemicals have a predicted difference within 25 Å${}^{2}$ by two approaches (shown in Figure 8). The lack of true CCS values in SusDat, thus, by comparing the differences in predicted results generated by two approaches, demonstrates the robustness of models, and the direct prediction model can discriminate different chemical classes.

Predicted CCS values are provided in Appendix A for use in non-target screening or retrospective analysis. Moreover, these predicted CCS values can be compared to the measured CCS values by standard inter-laboratory evaluation and inter-platform deviation and improve the performances of our models.

## 4. Discussion

In this study, we introduced topological fingerprints to categorize chemicals and generate CCS prediction models using the random forest algorithm. Our methods are generalized to TWIM and DTIM measured CCS data collected from seven sources. Prediction models were developed for five super classes of chemicals (Benzenoids, Lipids and lipid-like molecules, Organic acids and derivatives, Organic oxygen compounds, and Organoheterocyclic compounds) and the entire dataset. The test prediction accuracy was 0.958 by the direct prediction approach, 3 class-based prediction models more than 0.9, and over 0.86 for the remaining two classes. The MRE was between 1.89% to 2.33%. Additionally, models only required SMILES to encode fingerprints. A significant predicted variation was observed in macro molecules and vinyl acetate, with over 100 Å${}^{2}$ residual. We noticed that the residuals were reduced through the direct prediction model due to an extensive training set and a higher presence of macro molecules in the dataset. The prediction performances are highly dependent on the collected CCS libraries. Therefore, it is emphasized that multiple and accurate empirical CCS libraries with a broad scope of chemicals are crucial to CCS machine learning studies. Moreover, this bias indicated a limited prediction performance for chemicals with unique structures. A better classification model or other structural importance features might improve the prediction accuracy. Since fingerprint was the only input feature for prediction, adduct ions (e.g., [M + Na]^+^) were eliminated in this study. Other features can be introduced in the models to generate more ion types. Moreover, fingerprints offer a novel aspect in CCS prediction using machine learning. The generated feature importance of 1024 bits was directly related to the structures and thus easier to interpret chemically.

## Figures and Tables

**Figure 1 molecules-27-06424-f001:**
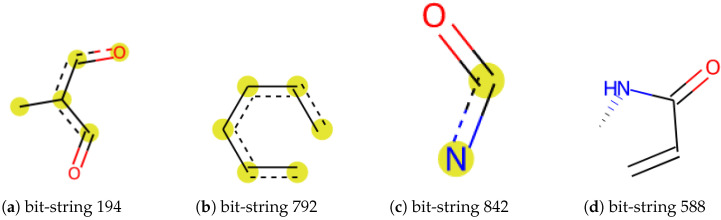
Examples of a substructure of a bit string. The most relevant features in the prediction models.

**Figure 2 molecules-27-06424-f002:**
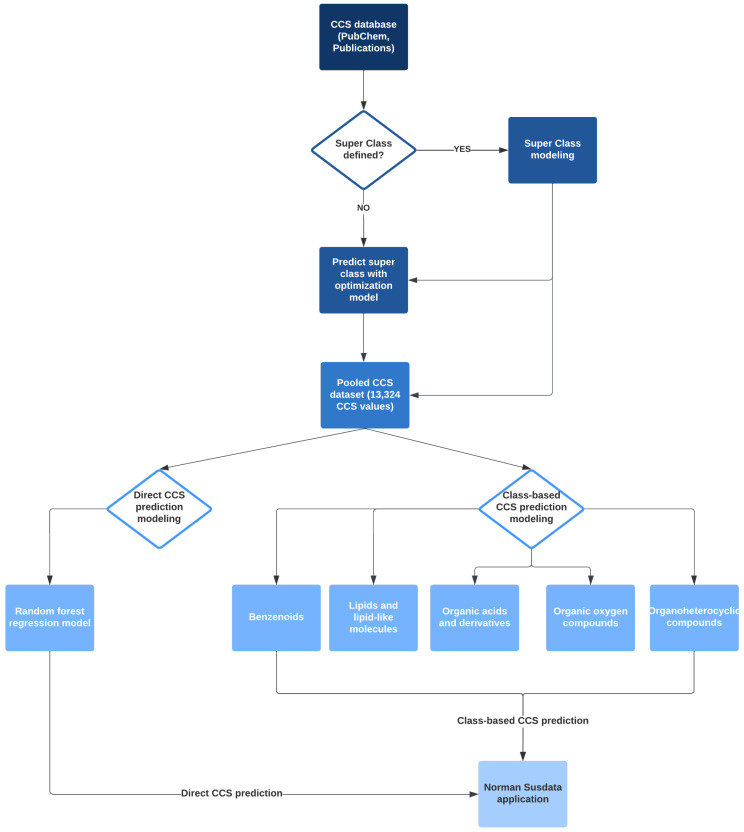
Modeling workflow: CCS empirical databases were collected from 6 different laboratories and PubChem. Two CCS prediction approaches were developed and validated. One model was class-based CCS prediction, and 5 super classes were defined for modeling. Another was a direct CCS prediction model. In the end, both prediction approaches were applied to the Norman Susdat list.

**Figure 3 molecules-27-06424-f003:**
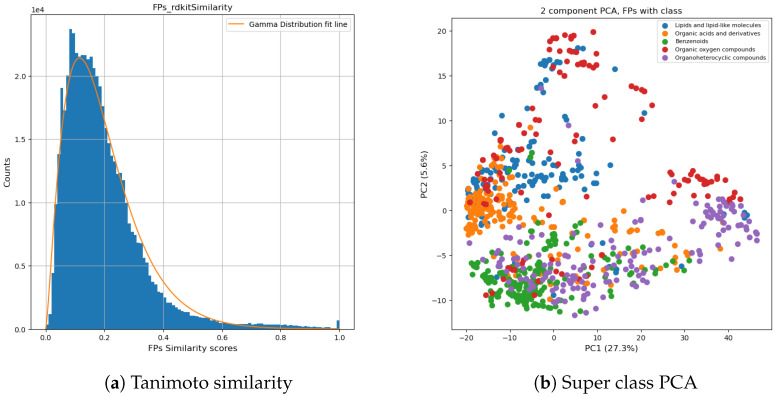
Super class distribution: A histogram of pair-wise fingerprints similarity is plotted in (**a**), and a normalized gamma distribution was fitted to the data and is shown as a red line. Based on the gamma distribution curve, similarity ≥ 0.6 was chosen to arrange the dataset. In (**b**), a 2D-scatter plot of PCA is generated by fingerprints.

**Figure 4 molecules-27-06424-f004:**
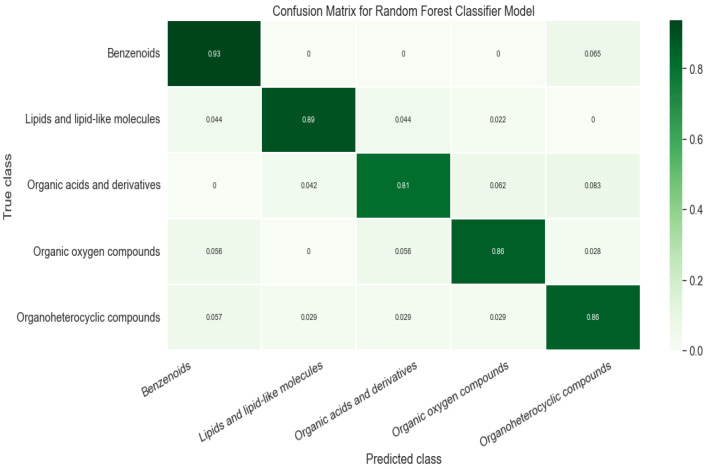
Confusion matrix of classification model.

**Figure 5 molecules-27-06424-f005:**
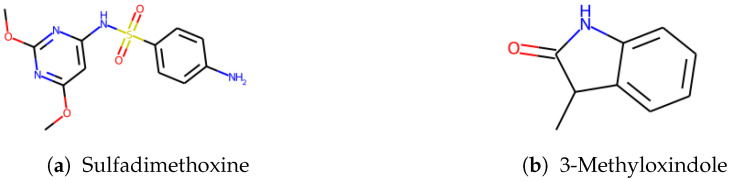
Random examples of “incorrect” predicted chemical.

**Figure 6 molecules-27-06424-f006:**
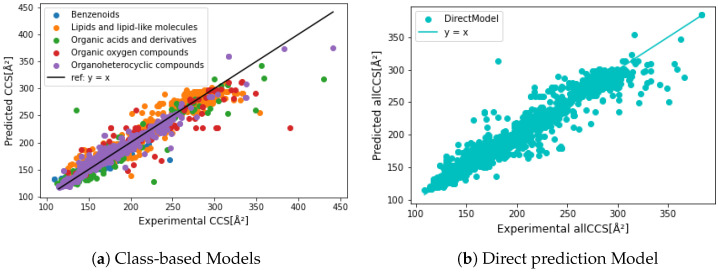
Precision comparison between each predictive class and direct prediction without class. Class-based models lead to a better precision from 150 to 300 Å${}^{2}$, while giving more bias by the small and macro molecules. The direct prediction model is less affected by the extreme cases.

**Figure 7 molecules-27-06424-f007:**
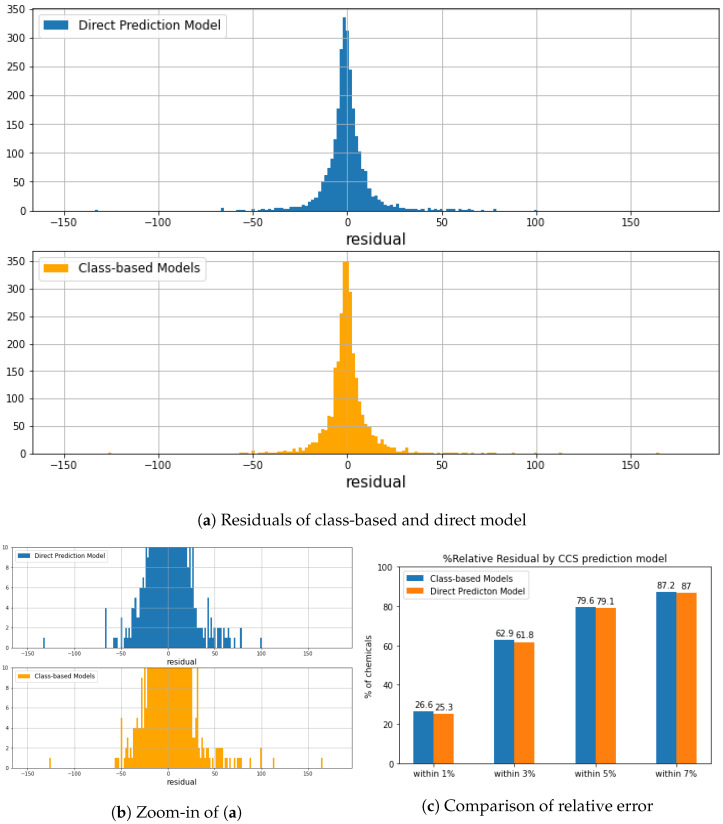
(**a**) compared the residuals of predicted CCS from class-based CCS prediction model and direct CCS prediction model. (**b**) is a zoomed-in of (**a**). Both approaches generate a good prediction power. A total of 98% of chemicals has a predicted difference within 25 Å${}^{2}$. (**c**) Comparison of relative error in the testing set between the two approaches within 1%, 3%, 5%, and 7%.

**Figure 8 molecules-27-06424-f008:**
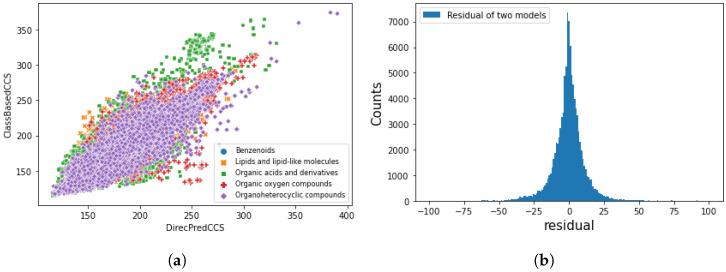
Comparison of direct and class-based CCS prediction model using Norman Susdat. (**a**) Scatter plot of class-based predicted CCS value against direct predicted CCS value. (**b**) Difference of predicted CCS values between class-based and direct prediction models. A total of 96 % of the chemicals have a predicted difference within 25 Å${}^{2}$.

**Table 2 molecules-27-06424-t002:** Results of super-class prediction modeling.

Super Class	Training	Test	F1 Score	Accuracy
Benzenoids	181	46	0.905	0.935
Lipids and lipid-like molecules	189	47	0.909	0.889
Organic acids and derivatives	184	46	0.848	0.813
Organic oxygen compounds	142	36	0.861	0.861
Organoheterocyclic compounds	140	35	0.822	0.857

**Table 3 molecules-27-06424-t003:** Results of CCS prediction modeling.

	Training		Test		
**Dataset**	**Data**	**R** ${}^{2}$	**Data**	**R** ${}^{2}$	**MRE (%)**
All	10,659	0.972	2665	0.958	2.20
Benzenoids	1930	0.942	483	0.869	1.89
Lipids and lipid-like molecules	3675	0.940	919	0.932	2.33
Organic acids and derivatives	1392	0.950	348	0.901	2.21
Organic oxygen compounds	754	0.925	189	0.860	2.33
Organoheterocyclic compounds	2907	0.960	724	0.933	1.96

## Data Availability

The datasets and the source codes can be found at https://github.com/fyang22/CCS-Prediction-Publish (accessed on 10 April 2022) and https://www.mdpi.com/article/10.3390/molecules27196424/s1.

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
