# Peer review of "Collision Cross Section Prediction with Molecular Fingerprint Using Machine Learning"

_molecules, 2022, doi:10.3390/molecules27196424_

Round 1

Reviewer 1 Report

Overall, the manuscript is interesting and the described collision cross-section prediction models seem useful. However, the language and the formatting, including the size and clarity of the figures, needs to be improved. The passages in the Introduction section referring to mass spectrometry are confusing and require substantial re-phrasing and clarification. The ML script together with the accompanying Git repository seems clear and reproducible.

·         The authors’ list in the SI contains affiliation indications, but no affiliations are provided.

·         The tiny sub-plots in Fig S4 overlap the captions. Please make Figs S2 and S4 more legible.

·         Line 1: I would argue that MS is a technique, not a methodology

·         Line 16: models

·         Line 25 and further: the structures of these compounds are unknown. I find the term “structurally unknown” confusing. Please rephrase.

·         Line 39: this line suggests that a fragmentation pattern can only be obtained through MS/MS and should be revised. Further, not all ToF and Orbitrap analysers are used in tandem MS (line 40); not all ToFs are QToFs. This passage is confusing.

·         line 42: it should be clarified that this passage refers to non-targeted analysis.

·         Line 123: why was the threshold set to 0.6? This is later explained in fig 3 caption but could be mentioned at this point.

·         Fig 3b: annotate the axes with the fraction of total variance explained by the first two PCs

·         Subplots in fig 7 are too small and thus not legible

·         in the Tuning section of the ClassificationModel the train_test_split is imported twice. Is there a reason for that?

·         Why did the authors choose RF in particular? If they tested different RF topologies, then why not test other supervised models?

·         It’s nice that the authors provide the trained models and implementation examples.

Author Response

Overall, the manuscript is interesting and the described collision cross-section prediction models seem useful. However, the language and the formatting, including the size and clarity of the figures, needs to be improved. The passages in the Introduction section referring to mass spectrometry are confusing and require substantial re-phrasing and clarification. The ML script together with the accompanying Git repository seems clear and reproducible.

We thank the reviewer for their positive evaluation of the manuscript. We also appreciate the fact that they tested our models and tools directly through our GitHub repo.

The authors’ list in the SI contains affiliation indications, but no affiliations are provided.

We thank the reviewer for this comment. These documents were prepared via Latex template provided by the publisher. This will be corrected during the type setting of the manuscript prior to publication.

The tiny sub-plots in Fig S4 overlap the captions. Please make Figs S2 and S4 more legible.

We thank the reviewer for this remark. The figures were updated with better resolution and layout.

Line 1: I would argue that MS is a technique, not a methodology

We have updated this in the revised version of the manuscript.

Line 16: models

We thank the reviewer for pointing out the typos and grammar issues in our manuscript. The revised version of the article has gone through another round of proofreading to avoid such issues.

Line 25 and further: the structures of these compounds are unknown. I find the term “structurally unknown” confusing. Please rephrase.

We thank the reviewer for their comment. We replaced ‘structurally unknown’ to ‘missing molecular and/or structure information’ in Line25, ‘unknown-unknown’ in Line33, ‘unknowns’ in Line 43.

Line 39: this line suggests that a fragmentation pattern can only be obtained through MS/MS and should be revised. Further, not all ToF and Orbitrap analysers are used in tandem MS (line 40); not all ToFs are QToFs. This passage is confusing.

We appreciate the corrections of the reviewer. this sentence was changed to (Line38-39):

HRMS (i.e. Time-of-flight (TOF) and Orbitrap) maintains a high mass accuracy within ± 5 mDa m/z error, and it can be acquired in full scan MS data or plus MS/MS data. 

line 42: it should be clarified that this passage refers to non-targeted analysis.

We have clarified this in the revised manuscript.

Line 123: why was the threshold set to 0.6? This is later explained in fig 3 caption but could be mentioned at this point.

This has been addressed in the revised manuscript in line130-133.

‘Chemicals in other super class were assigned to one of the referred classes based on their similarity with a minimum similarity threshold of 0.6 since around 97\% of pair-wise similarities were under 0.6 (shown in Figure 3)’

Fig 3b: annotate the axes with the fraction of total variance explained by the first two PCs

We thank the reviewer for their remark, the figure has been updated with the fraction of total variance.

Subplots in fig 7 are too small and thus not legible

We thank the reviewer for their suggestion. We have improved the layout of fig 7.

in the Tuning section of the ClassificationModel the train_test_split is imported twice. Is there a reason for that?

We thank the reviewer for pointing this out. We have rectified this in the revised script and uploaded it in our GitHub repo.

Why did the authors choose RF in particular? If they tested different RF topologies, then why not test other supervised models?

We choose RF because it is one of the common supervised models for nonlinear and categorical data. Given the excellent results obtained with random forest, we did not see any justification for further model testing.

It’s nice that the authors provide the trained models and implementation examples.

We appreciate the positive feedback of the reviewer, regarding our open science practices.

Reviewer 2 Report

The manuscript ‘Collision Cross Section Prediction with Molecular Fingerprint Using Machine Learning’ by Yang at al. reports an interesting study describing a new machine learning approach for predicting Collision Cross Section (CCS). More precisely the main novelty of the present study is the use of molecular topology fingerprint instead of molecular descriptors for carrying out the whole procedure. The manuscript is well written and the reported study, by a technical point of view is very well performed. I do not have specific remarks. On the other hand it is not particularly clear, to a non-expert reader potentially interested to this work, what are the actual advantages of using the molecular fingerprint – instead of the (probably) more physical and general descriptors – from which the user can benefit. Is it only a matter of ease of access to the information? I only have a further personal (very general) concern: the authors at p. 2 claim that ‘CCS values can be measured accurately with a deviation < 3%. I am a bit skeptical, for personal experience, about this statement. Probably the authors are right if ther are referring to an estimation of the standard error of a series of measurements from the same lab. On the other hand I wouldn’t be so optimistic.

Author Response

The manuscript ‘Collision Cross Section Prediction with Molecular Fingerprint Using Machine Learning’ by Yang at al. reports an interesting study describing a new machine learning approach for predicting Collision Cross Section (CCS). More precisely the main novelty of the present study is the use of molecular topology fingerprint instead of molecular descriptors for carrying out the whole procedure. The manuscript is well written and the reported study, by a technical point of view is very well performed. I do not have specific remarks.

We very much appreciate the reviewer’s positive feedback about the novelty and importance of our work.

On the other hand it is not particularly clear, to a non-expert reader potentially interested to this work, what are the actual advantages of using the molecular fingerprint – instead of the (probably) more physical and general descriptors – from which the user can benefit. Is it only a matter of ease of access to the information?

The molecular descriptors were selected for this study, given that they are as accurate as 0D and 1D descriptors. At the same time, they incorporate additional structural information at 2D and 3D levels, without the need for the structural optimization. We have provided additional description of the fingerprints in the revised manuscript.

 I only have a further personal (very general) concern: the authors at p. 2 claim that ‘CCS values can be measured accurately with a deviation < 3%. I am a bit skeptical, for personal experience, about this statement. Probably the authors are right if their are referring to an estimation of the standard error of a series of measurements from the same lab. On the other hand I wouldn’t be so optimistic.

Firstly, we thank the reviewer pointed out our unclear argument. It was clarified in the manuscript in line 53-56:

 Studies have demonstrated that the inter-laboratory CCS biases are within 2% for the same IMS technique [36,37]. Moreover, cross-platform biases are below 3% for over 98% of the chemicals included in their studies [38, 39].

According to our bibliography research (with hyperlinks below), if the external calibration is correctly performed, the bias of CCS within the same IMS technique was within 1.5% (https://doi.org/10.1021/acs.analchem.7b01729, https://doi.org/10.1021/acs.analchem.9b05247).

And for different IMS techniques (TWIM, DTIM and TIMS), CCS bias are within 3%. There are less than 2% of the chemicals had a cross-platform CCS bias up to 6 or 7 %, but we considered them as specific cases (https://doi.org/10.1021/jasms.2c00196, https://doi.org/10.1021/acs.analchem.8b02711, https://doi.org/10.1007/s00216-022-04263-5, https://doi.org/10.1021/jasms.1c00056).

Reviewer 3 Report

The manuscript by Yang et al. proposes machine learning models for CCS prediction using molecular topology fingerprints. The methods for superclass classification and CCS value predictions are solid, and the predicted results reach reasonable metrics. The reviewer considers the paper can be published on Molecules after addressing the following questions.

1. The machine learning approach proposed by the paper strongly depends on the embedding quality of molecular topology fingerprints by the external library RDKit. The authors should include more details about the used embedding method in the method section.

2. The test results of the direct method are better than the class-based method; however, the reason is not well discussed except for the dataset size difference. The reviewer recommends the authors discuss whether it can be understood from the derived feature importance. 

Author Response

The manuscript by Yang et al. proposes machine learning models for CCS prediction using molecular topology fingerprints. The methods for superclass classification and CCS value predictions are solid, and the predicted results reach reasonable metrics. The reviewer considers the paper can be published on Molecules after addressing the following questions.

We thank the reviewer for the positive evaluation of our manuscript.

The machine learning approach proposed by the paper strongly depends on the embedding quality of molecular topology fingerprints by the external library RDKit. The authors should include more details about the used embedding method in the method section.

We understand and appreciate reviewer’s comment. Indeed, any machine learning algorithm is highly dependent on the quality of its training set. We have provided our reasoning behind the use of fingerprints in our response to reviewer 2. Additionally, we have added the below text to the manuscript (in line 97-102) to further clarify this:

Topological torsion (TT) fingerprints were chosen as features to encode chemical structure. TT fingerprints were first introduced by Nilakantan et al., which describe the atom type, the topological distance between two atoms within four bonds, and torsion angles [56]. Four examples of molecular substructures were shown in Figure 1. The SMILES were converted to 1024 bit-strings fingerprints (FPs) by implemented module in RDKit.

The test results of the direct method are better than the class-based method; however, the reason is not well discussed except for the dataset size difference. The reviewer recommends the authors discuss whether it can be understood from the derived feature importance.

We understand the reviewer’s comment related to the observed differences between the two model sets. For both model sets, the largest errors are observed for the extreme cases, which have the smallest fraction of the training set. This was a lot more pronounced in the class-based models, given the size of the training set. At the current state and with the available size of the training set, we believe in further chemical conclusions (i.e. through feature importance) may not be generalizable. Therefore, we limited our explanation to the size of the training set.

Round 2

Reviewer 1 Report

I believe that the Editor can consider the revised version of the manuscript for publication.